# D2RL: Deep Dense Architectures in Reinforcement Learning

## Abstract

While improvements in deep learning architectures have played a crucial role in improving the state of supervised and unsupervised learning in computer vision and natural language processing, neural network architecture choices for reinforcement learning remain relatively under-explored. We take inspiration from successful architectural choices in computer vision and generative modeling, and investigate the use of deeper networks and dense connections for reinforcement learning on a variety of simulated robotic learning benchmark environments. Our findings reveal that current methods benefit significantly from dense connections and deeper networks, across a suite of manipulation and locomotion tasks, for both proprioceptive and image-based observations. We hope that our results can serve as a strong baseline and further motivate future research into neural network architectures for reinforcement learning. The project website is at this link https://sites.google.com/view/d2rl-anonymous/home

## 1 Introduction

Deep Reinforcement Learning (DRL) is a general purpose framework for training goal-directed agents in high dimensional state and action spaces. There have been plenty of successes from DRL for robotic control tasks, spanning across locomotion and navigation tasks, both in simulation and in the real world (Schulman et al., 2015; Akkaya et al., 2019; Kalashnikov et al., 2018).

While the generality of the DRL framework lends itself to be applicable to a wide variety of tasks, one has to address issues such as the sample-efficiency and generalization of the agents trained with this framework. Sample-efficiency is fundamentally critical to agents trained in the real world, particularly for robotic control tasks. Baking in *minimal* inductive biases into the framework is one effective mechanism to address the issue of sample-efficiency of DRL agents and make them more efficient.

The generality of the framework makes it difficult to control particular behaviours and inductive biases for DRL algorithms. Inductive biases are important for learning algorithms, as they are able to induce desirable behaviour in the learned agents. Recent work has sought to improve the sample efficiency of DRL by adding an inductive bias of invariance, when learning from images, through techniques such as data augmentations (Laskin et al., 2020; Kostrikov et al., 2020) and contrastive losses (Srinivas et al., 2020). Similarly, another important inductive bias in DRL is the choice of the architectures for function approximators, for example how to parameterize the neural network for the policy and value functions. However, the problem of choosing architecture designs in DRL and robotics, for planning and control, has been largely ignored.

Modern computer vision and language processing research have shown the disproportionate advantage of the size and depth of the neural networks used (He et al., 2016b; Radford et al.) wherein *very* deep neural networks can be trained such that they learn better and more generalizable representations. Furthermore, recent evidence suggests that deeper neural networks can not only learn more complex functions but also have a smoother loss landscape (Rolnick & Tegmark, 2017). Learning function approximators which enable better optimization and expressivity is an important inductive bias, which is greatly exploited in vision and language processing by using clever neural network architecture choices such as residual connections (He et al., 2016a), normalization layers (Santurkar et al., 2018), and gating mechanisms (Hochreiter & Schmidhuber, 1997), to name a few. It would be ideal to incorporate similar inductive biases in modern DRL algorithms in robotics in order to

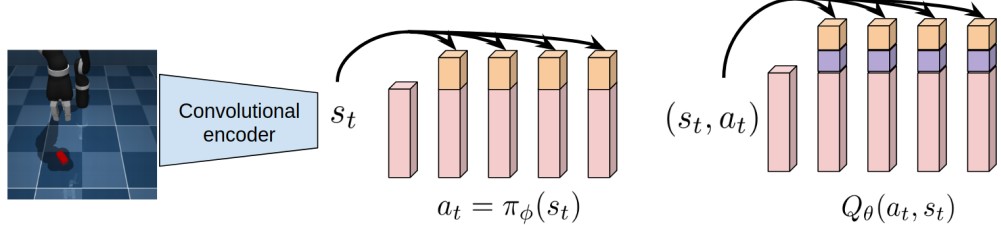

**Figure 1:** Visual illustrations of the proposed dense-connections based `D2RL` modification to the policy $\pi_\phi(\cdot)$ and Q-value $Q_\theta(\cdot)$ neural network architectures. The inputs are passed to each layer of the neural network through identity mappings. Forward pass corresponds to moving from left to right in the figure. For state-based envs, $s_t$ is the observed simulator state and there is no convolutional encoder.

allow for better sample efficiency as that would significantly aid the deployment of real world robot learning agents.

In this paper, we first highlight the problems that occur when learning policies and value functions using vanilla deep neural networks. Then we propose `D2RL`; an architecture that addresses these problems while benefiting from the utility of inductive biases added by more expressive function approximators. We show how our proposed architecture scales effectively to a wide variety of off-policy RL algorithms, for proprioceptive-feature and image based inputs across a diverse set of challenging robotic control and manipulation environments. Our approach is motivated by utilizing a form of dense-connections similar to the ones found in modern deep learning, such as DenseNet (Huang et al., 2017), Skip-VAE (Dieng et al., 2019) and U-Nets (Ronneberger et al., 2015). We demonstrate that the proposed parameterization significantly improves sample efficiency of RL agents (as measured by the number of environment interactions required to obtain a level of performance) in continuous control tasks.

Our contributions can be summarized as:

1. We investigate the problem with increasing the number of layers used to parameterize policies and value functions.
2. We propose a general solution based on dense-connections to overcome the problem.
3. We extensively evaluate our proposed architecture on a diverse set of robotics tasks from proprioceptive features and images across multiple standard algorithms.

## 2   RELATED WORK

**Learning efficient representations for sample-efficient RL**   Several recent papers have sought to improve representation learning of observations for control. CURL (Srinivas et al., 2020) augments the usual RL loss with a contrastive loss that seeks to learn a latent representation which enforces the inductive bias of encodings of augmentations of the same image being closer in latent space than embeddings of augmentations of different images. RAD (Laskin et al., 2020) and DrQ (Kostrikov et al., 2020) showed that simple data augmentations like random crop, color jitter, patch cutout, and random convolutions can alone help improve sample-efficiency and generalization of RL from pixel inputs. Some other algorithms learn latent representations decoupled from policy learning, through a variational autoencoder based loss (Higgins et al., 2017; Hafner et al., 2019; Nair et al., 2018). OFENet (Ota et al., 2020) shows that learning a higher dimensional feature space helps learn a more informative representation when learning from states.

**Inductive biases in deep learning**   Inductive biases in deep learning have been long explored in different contexts such as temporal relations (Hochreiter & Schmidhuber, 1997), spatial relations (LeCun et al., 1998; Krizhevsky et al., 2012), translation invariance (Berthelot et al., 2019; He et al., 2020; Chen et al., 2020; Srinivas et al., 2020; Laskin et al., 2020) and learning contextual representations (Vaswani et al., 2017). These inductive biases are injected either directly through the network parameterization (LeCun et al., 1998; Hochreiter & Schmidhuber, 1997; Vaswani et al., 2017) or by implicitly changing the objective function (Berthelot et al., 2019; Srinivas et al., 2020).

**Learning very deep networks** Deep neural networks are useful for extracting features from data relevant for various downstream tasks. However, simply increasing the depth of a feed-forward neural network leads to instability in training due to issues such as vanishing gradients (Hochreiter & Schmidhuber, 1997), or a a loss of mutual information (He et al., 2016a). To mitigate this, residual connections were proposed which involve an alternate path between layers through an identity mapping (He et al., 2016a). Skip-VAEs (Dieng et al., 2019) tackle a similar issue of posterior collapse in typical VAE training by adding skip connections in the architecture of the VAE decoder. U-Nets (Ronneberger et al., 2015) consider a contractive path of convolutions and maxpooling layers followed by an expansive path of up-convolution layers. There are copy and crop identity mapping from layers of the contractive path to layers in the expansive path. Normalization techniques such as batch normalization are also important in learning deep networks (Ioffe & Szegedy, 2015). Combining residual connections with batch normalization have been used to successfully train networks with 1000 layers (He et al., 2016b). Our proposed architecture closely resembles DenseNet (Huang et al., 2017), which uses skip connections from feature maps of previous layers through concatenation, allowing for efficient learning and inference.

## 3 PRELIMINARIES

In this section, we describe the actor-critic formulation of RL algorithms, which serves as the basic framework for our setup. We then describe the Data-Processing Inequality which is relevant for explaining and motivating the proposed architecture.

### 3.1 ACTOR-CRITIC METHODS

Actor-critic methods learn two models, one for the policy function and the other for the value function such that the value function assists in the learning of the policy. These are TD-learning (Tesauro, 1995) methods that have an explicit representation for the policy independent of the value function. We consider the setting where the critic is an state-action value function $Q_\theta(a, s)$ parameterized by a neural network, and the actor $\pi_\phi(a|s)$ is also parameterized by a neural network. Let the current state be $s$ and $r_t$ denote the reward obtained after executing action $a$ in state $s$, and transitioning to state $s'$. After sampling action $a' \sim \pi_\phi(a'|s)$ in the next state $s'$, the policy parameters are updated in the direction suggested by the critic $Q_\theta(a, s)$

$$\phi \leftarrow \phi + \beta_\phi Q_\theta(a, s)\nabla_\phi \log \pi_\phi(a|s).$$

The parameters $\theta$ are updated using the TD correction $\Delta_t = r_t + \gamma Q_\theta(s', a') - Q_\theta(s, a)$ as follows:

$$\theta \leftarrow \theta + \beta_\theta \Delta_t \nabla_\theta Q_\theta(s, a).$$

Although the basic formulation above requires on-policy samples $(s, a, s', r)$ for gradient updates, a number of off-policy variants of the algorithm have been proposed (Haarnoja et al., 2018; Fujimoto et al., 2018) that incorporate importance weighting in the policy's gradient update. Let the observed samples be sampled by a behavior policy $a \sim \zeta(a|s)$, and $\pi_\phi(a|s)$ be the policy to be optimized. The gradient update rule changes as

$$\phi \leftarrow \phi + \beta_\phi \frac{\pi_\phi(a|s)}{\zeta(a|s)} Q_\theta(a, s)\nabla_\phi \log \pi_\phi(a|s)$$

### 3.2 DATA-PROCESSING INEQUALITY

The Data processing inequality (DPI) states that the information content of a signal cannot be increased via a local physical operation. So, given a Markov chain $X_1 \rightarrow X_2 \rightarrow X_3$, the mutual information (MI) between $X_1$ and $X_2$ is not less than the MI between $X_1$ and $X_3$ i.e.

$$MI(X_1; X_2) \geq MI(X_1; X_3)$$

A vanilla feed-forward neural network has successive layers depend only on the output of the previous layer and so there is a Markov chain of the form $X_1 \rightarrow X_2 \rightarrow \cdots \rightarrow X_n$ from the input $X_1$ to the final output $X_n$. In practice the last layer $X_n$ contains *less information* than the previous layer $X_{n-1}$. By using dense connections (Huang et al., 2017), we are able to overcome the problem of DPI, as the original input is simply concatenated with the intermediate layers of the networks. We postulate that using dense connections are also important when parameterizing policies and value networks in RL and robotics. By preserving important information about the input across layers explicitly through dense connections, we can achieve faster convergence in solving complex tasks.

### 3.3 IMPLICIT UNDER-PARAMETERIZATION IN DEEP Q-LEARNING

Kumar et al. (2020) showed that using MLPs for function approximation of the policy and value functions in deep RL algorithms that use bootstrapping, leads to an implicit-underparameterization phenomena that causes poor-er performance. Implicit under-parameterization refers to a reduction in the effective rank of the feature, $srank_\delta(\Phi)$ that occurs implicitly due to using MLPs for approximating Q-functions. This causes rank collapse for the feature matrix $\Phi$ which are the weights of the penultimate layer of the Q network during training.

We believe that adding skip connections to the network architecture, as in D2RL could help alleviate this rank collapse issue, and improve performance. We empirically verify this in Section 5.2 and Table 3.

## 4 METHOD

In this section, we first show the issues with using deeper Multi-layered Perceptons (MLPs) to parameterize the policies and Q-networks in RL and robotics due to the Data Processing Inequality (DPI). We then propose a simple and effective architecture which overcomes the issues, using dense-connections. We will denote our proposed method as: Deep Dense architectures for Reinforcement Learning or `D2RL` in the subsequent sections.

### 4.1 PROBLEM WITH DEEPER NETWORKS IN REINFORCEMENT LEARNING

To test whether we can use the expressive power of deeper networks in RL for robotic control, we train a SAC agent (Haarnoja et al., 2018), while increasing the number of layers in the MLP used to parameterize the policy and the Q-networks, on the Ant-v2 environment of the OpenAI Gym suite (Brockman et al., 2016). The results are shown in Fig. 2. The results suggest that by simply increasing the number of layers used to parameterize the networks, we are unable to benefit from the inductive bias of deeper feature extractors in RL, like we see in computer vision. As we increase the number of layers in the network, the mutual information between the output and input likely decreases due to the non-linear transformations used in deep learning as explained by the DPI (see Section 3.2). Increasing the number of layers from 2 to 4 significantly decreases the sample efficiency of the agent and furthermore, increasing the number of layers from 2 to 8 decreases the sample efficiency while also making the agent less stable during training. As expected, when instead decreasing the number of layers from 2 to 1 the vanilla MLPs are not sufficiently expressive to perform well on the Ant-v2 task. This simple experiment suggests that even though the expressivity of the function approximators is important, by simply adding more layers in a vanilla MLP, the agent's performance and sample complexity significantly deteriorates. However, it is desirable to use deeper networks to increase the network expressivity and to enable better optimization for the networks.

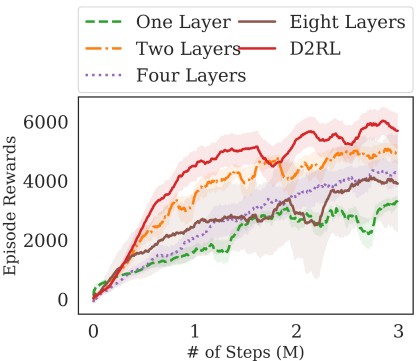

**Figure 2:** The effect of increasing the number of fully-connected layers to parameterize the policy and Q-Networks for Soft-Actor Critic (Haarnoja et al., 2018) on Ant-v2 in the OpenAI Gym Suite (Brockman et al., 2016). It is evident that performance drops when increasing depth after 2 layers. However, our `D2RL` agent with 4 layers does not suffer from this, and performs better.

### 4.2 D2RL

Our proposed `D2RL` variant incorporates dense connections (input concatenations) from the input to each of the layers of the MLP used to parameterize the policy and value functions. We simply concatenate the state or the state-action pair to each hidden layer of the networks except the last output linear layer, since that is just a linear transformation of the output from the previous layer. In case of pixel observations, we consider the states to be the encodings of a CNN encoder, as shown in Fig. 1. Our simple architecture enables us to increase the depth of the networks while also satisfying DPI. Fig 1 is a visual illustration and we provide a PyTorch-like pseudo-code below to promote

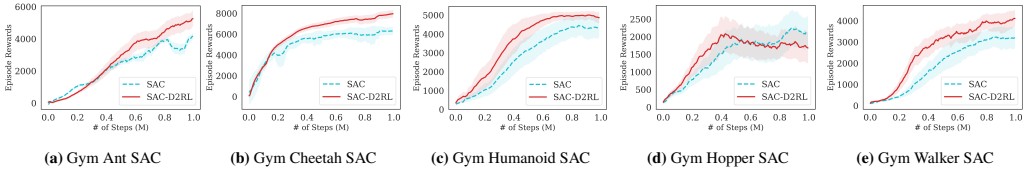

**(a)** Gym Ant SAC      **(b)** Gym Cheetah SAC      **(c)** Gym Humanoid SAC      **(d)** Gym Hopper SAC      **(e)** Gym Walker SAC

**Figure 3: OpenAI Gym benchmark environments with SAC.** Comparison of the proposed `D2RL` and the baselines on a suite of OpenAI-Gym environments. We apply the `D2RL` modification to SAC (Haarnoja et al., 2018). The error bars are with respect to 5 random seeds. The results on Humanoid env are in the Appendix.

clarity of the proposed method. We also include the actual PyTorch code (Paszke et al., 2019) for the policy and value networks to allow for fast-adoption and reproducibility in Appendix A.

```
1  # Sample state, action from the replay buffer
2  state, action = replay_buffer.sample()
3  # Feed state, action into the first linear layer of a Q-network
4  q_input = concatenate(state, action)
5  h = MLP(q_input)
6  # Concatenate the hidden representation with the input
7  h = concatenate(h, q_input)
8  # Feed the concatenated representation into the second layer
9  h = MLP(h)
```

## 5 EXPERIMENTS

We experiment across a suite of different environments, some of which are shown in Fig. 8, each simulated using the MuJoCo simulator. We were unable to do real robot experiments due to COVID-19 and have included a 1-page statement along with the Appendix describing how our method can conveniently scale to physical robots. Through the experiments, we aim to understand the following questions:

- How does `D2RL` compare with the baseline algorithms in terms of both asymptotic performance and sample efficiency on challenging robotic environments?
- Are the benefits of `D2RL` consistent across a diverse set of algorithms and environments?
- Is `D2RL` important for both the policy and value networks? How does `D2RL` perform with increasing depth?

### 5.1 EXPERIMENTAL DETAILS

To enable fair comparison between the current standard baselines and `D2RL`, we simply replace the 2-layer MLPs that are commonly parameterize the policy and value function(s) in widely used actor-critic algorithms such as SAC (Haarnoja et al., 2018; Yarats et al., 2019), TD3 (Fujimoto et al., 2018), DDPG (Qiu et al., 2019) and HIRO (Nachum et al., 2018). Instead, we use 4-layer `D2RL` to parameterize both the policies and the value function(s) in each of the actor-critic algorithms. Outside of the network architecture, we **do not change any hyperparameters**, and use the exact same values as reported in the original papers and the corresponding open-source code repositories. The details of all the hyperparameters used are in Appendix C. We perform ablation studies in Section 5.4 to $i$) investigate the importance of parameterizing both the policy and value function(s) using `D2RL` and $ii$) see how varying the number of layers of `D2RL` affects performance. We also perform further experiments with a ResNet style architecture and additional experiments with Hindsight Experience Replay (Andrychowicz et al., 2017) on simpler manipulation environments which can be found in Appendix B.

### 5.2 EXPERIMENTAL RESULTS

**The proposed `D2RL` variant achieves superior performance compared to the baselines on state-based OpenAI Gym MuJoCo environments.** We benchmark the proposed `D2RL` variant on a suite of OpenAI Gym (Brockman et al., 2016) environments by applying the modification to two standard

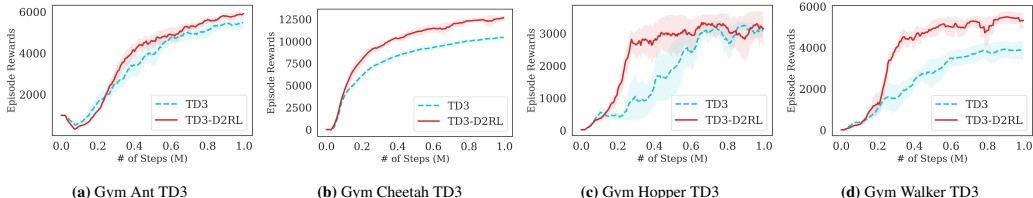

**(a)** Gym Ant TD3     **(b)** Gym Cheetah TD3     **(c)** Gym Hopper TD3     **(d)** Gym Walker TD3

**Figure 4: OpenAI Gym benchmark environments with TD3.** Comparison of the proposed variation D2RL and the baselines on a suite of OpenAI-Gym environments. We apply the D2RL modification to TD3 (Fujimoto et al., 2018). The error bars are with respect to 5 random seeds.

| | From Images | | | From States | | |
|---|---|---|---|---|---|---|
| **100K Step score** | **CURL** | **CURL-ResNet** | **CURL-D2RL** | **SAC** | **SAC-ResNet** | **SAC-D2RL** |
| Finger, Spin | $767 \pm 57$ | $548 \pm 120$ | $\mathbf{837} \pm 18$ | $459 \pm 48$ | $551 \pm 57$ | $\mathbf{627} \pm 107$ |
| Cartpole, Swing | $582 \pm 142$ | $327 \pm 101$ | $\mathbf{836} \pm 34$ | $717 \pm 14$ | $701 \pm 24$ | $\mathbf{751} \pm 12$ |
| Reacher, Easy | $538 \pm 233$ | $526 \pm 79$ | $\mathbf{754} \pm 168$ | $\mathbf{752} \pm 121$ | $626 \pm 112$ | $675 \pm 203$ |
| Cheetah, Run | $\mathbf{299} \pm 48$ | $230 \pm 17$ | $253 \pm 57$ | $587 \pm 58$ | $633 \pm 28$ | $\mathbf{721} \pm 43$ |
| Walker, Walk | $403 \pm 24$ | $275 \pm 57$ | $\mathbf{540} \pm 143$ | $132 \pm 43$ | $\mathbf{456} \pm 89$ | $354 \pm 159$ |
| Ball in Cup, Catch | $769 \pm 43$ | $450 \pm 169$ | $\mathbf{880} \pm 48$ | $867 \pm 42$ | $\mathbf{880} \pm 22$ | $\mathbf{891} \pm 33$ |
| **500K Step score** | **CURL** | **CURL-ResNet** | **CURL-D2RL** | **SAC** | **SAC-ResNet** | **SAC-D2RL** |
| Finger, Spin | $926 \pm 45$ | $896 \pm 59$ | $\mathbf{970} \pm 14$ | $899 \pm 29$ | $917 \pm 21$ | $\mathbf{961} \pm 8$ |
| Cartpole, Swing | $841 \pm 45$ | $833 \pm 9$ | $\mathbf{859} \pm 8$ | $884 \pm 2$ | $885 \pm 4$ | $885 \pm 2$ |
| Reacher, Easy | $\mathbf{929} \pm 44$ | $900 \pm 48$ | $\mathbf{929} \pm 62$ | $\mathbf{973} \pm 23$ | $969 \pm 34$ | $952 \pm 30$ |
| Cheetah, Run | $\mathbf{518} \pm 28$ | $459 \pm 108$ | $386 \pm 115$ | $781 \pm 65$ | $807 \pm 69$ | $\mathbf{842} \pm 75$ |
| Walker, Walk | $902 \pm 43$ | $807 \pm 134$ | $\mathbf{931} \pm 24$ | $959 \pm 16$ | $\mathbf{972} \pm 8$ | $964 \pm 14$ |
| Ball in Cup, Catch | $959 \pm 27$ | $960 \pm 8$ | $955 \pm 15$ | $976 \pm 12$ | $970 \pm 19$ | $972 \pm 13$ |

**Table 1: DeepMind control suite *benchmark* environments from images (CURL) and states (SAC).** Results of CURL (Srinivas et al., 2020), CURL-D2RL, SAC (Haarnoja et al., 2018), and SAC-D2RL, on the standard DM Control Suite *benchmark* environments. CURL (Srinivas et al., 2020) and CURL-D2RL are trained purely with pixel observations while SAC (Haarnoja et al., 2018) and SAC-D2RL are trained with proprioceptive features. The results for CURL were taken directly as reported by Srinivas et al. (2020). CURL-ResNet is the baseline that uses a similar network as CURL-D2RL but with residual connections. Similarly, SAC-ResNet is the baseline that uses a similar network as SAC-D2RL but with residual connections. The S.D. is over 5 random seeds.

| **100K Step Score** | **TD3** | **TD3-D2RL** | **500K Step Score** | **TD3** | **TD3-D2RL** |
|---|---|---|---|---|---|
| Walker, Walk-Easy | $\mathbf{67} \pm 8$ | $\mathbf{69} \pm 11$ | Walker, Walk-Easy | $\mathbf{79} \pm 10$ | $77 \pm 8$ |
| Walker, Walk-Medium | $38 \pm 8$ | $\mathbf{60} \pm 9$ | Walker, Walk-Medium | $44 \pm 8$ | $\mathbf{69} \pm 6$ |
| Cartpole, Swing-Easy | $96 \pm 24$ | $\mathbf{142} \pm 18$ | Cartpole, Swing-Easy | $102 \pm 24$ | $\mathbf{171} \pm 32$ |
| Cartpole, Swing-Medium | $67 \pm 2$ | $\mathbf{102} \pm 13$ | Cartpole, Swing-Medium | $68 \pm 2$ | $\mathbf{138} \pm 16$ |

**Table 2: Real World RL suite environments from states.** Results of TD3 (Haarnoja et al., 2018), and TD3-D2RL, on the Real World RL suite environments after 500K environment steps over 5 seeds. We see that using D2RL we are able to perform better in environments with distractors, random noise and delays. These experiments show how D2RL is able to learn robust agents.

RL algorithms, SAC (Haarnoja et al., 2018) and TD3 (Fujimoto et al., 2018). For all the environments, namely Ant, Cheetah, Hopper, Humanoid, and Walker, the observations received by the agent are state vectors consisting of the positions and velocities of joints of the agent. Additional details about the state-space and action-space are in the Appendix. From Fig. 4, we see that the proposed D2RL modification converges to significantly higher episodic rewards in most environments, and in others is competitive with the baseline. Also, these benefits can be seen across both the algorithms, SAC and TD3.

**D2RL is more sample efficient compared to baseline state-of-the-art algorithms on both image-based and state-based DM Control Suite *benchmark* environments.** We compare the proposed D2RL variant with SAC and state-of-the-art pixel-based CURL algorithms on the benchmark envi-

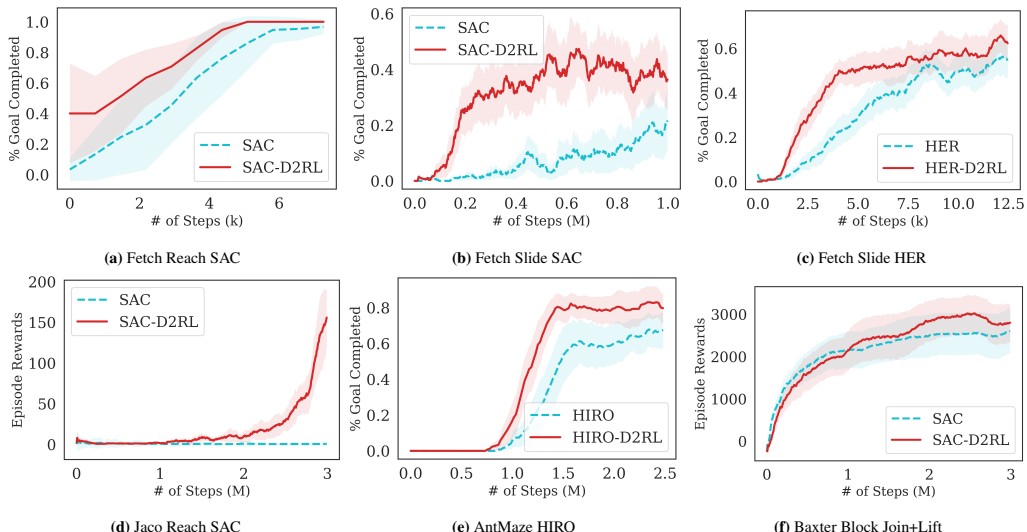

**Figure 5: Challenging selected manipulation and locomotion environments.** Comparison of the proposed variation D2RL and the baselines on a suite of challenging manipulation and locomotion environments. We apply the D2RL modification to the SAC (Haarnoja et al., 2018), HER (Andrychowicz et al., 2017), and HIRO (Nachum et al., 2018) algorithms and compare relative performance in terms of average episodic rewards with respect to the baselines. The task complexity increases from Fetch Reach to Fetch Slide. Jaco Reach is challenging due to high-dimensional torque controller action space, AntMaze requires exploration to solve a temporally extended problem, and Furniture BlockJoin requires solving two tasks- join and lift sequentially. The error bars are with respect to 5 random seeds. Some additional results on the Fetch envs are in the Appendix.

ronments of DeepMind Control Suite (Tassa et al., 2020). For CURL, and CURL-D2RL, we train using only pixel-based observations from the environment. For SAC and SAC-D2RL, we train using proprioceptive features from the environment. The action spaces are the same in both the cases, pixels and state features based observations. The environments we consider are part of the *benchamrk suite*, and include Finger Spin, Cartpole Swing, Reacher Easy, Cheetah Run, Walker Walk, Ball in Cup Catch. Additional details are in the Appendix.

In Table 1, we tabulate results for all the algorithms after 100K environment interactions, and after 500K environment interactions. To report the results of the baseline, we simply use the results as reported in the original paper (Srinivas et al., 2020). From this Table, we observe that the D2RL variant performs better than the baselines for both 100K and 500K environment interactions, and the performance gains are especially significant in the 100K step scores. This indicates that D2RL is significantly more sample-efficient than the baseline.

**D2RL performs significantly better in challenging environments with various modalities of noise, system delays, physical perturbations and dummy-dimensions (Dulac-Arnold et al., 2020).** Dulac-Arnold et al. (2019) propose a set of challenges in the DM Control Suite environments that are more "realistic" by introducing the aforementioned problems. We train a TD3 agent Fujimoto et al. (2018) from states, on the "easy" and "medium" challenges for the walker-walk, and cartpole-swingup environments with and without D2RL. We present the results in Table 2. We see how the baseline TD3 agent gets significantly worse in the "medium" challenge compared to the "easy" version of the same environment. The agent trained with TD3-D2RL significantly outperforms the baseline TD3 agent on 3 of the 4 challenges, and the drop between the "easy" and "medium" challenges is significantly less severe, compared to the baseline. This experiment demonstrates how by using D2RL, we are able to get significantly better performance on environments which have been constructed to be more realistic by adding difficult problems that the agent must learn to reason with. The increased robustness to such problems further validates the general utility of D2RL in many different circumstances.

**Verifying the alleviation of implicit underparameterization with D2RL.** In Table 3, we compare effective ranks of the feature matrices for D2RL and a normal MLP, with the TD3 algorithm after 1M interactions, corresponding to Fig. 4. We observe that the effective ranks for D2RL are higher

| 1M Steps | Walker2d-v2 | | Ant-v2 | | Hopper-v2 | | Cheetah-v2 | |
|---|---|---|---|---|---|---|---|---|
| | TD3 | TD3-D2RL | TD3 | TD3-D2RL | TD3 | TD3-D2RL | TD3 | TD3-D2RL |
| Policy | 153 | **161** | 165 | **178** | 145 | **159** | 157 | **161** |
| Q-network | 138 | **165** | 153 | **180** | 123 | **157** | 143 | **164** |

**Table 3: OpenAI Gym benchmark environments with TD3.** Comparison of the proposed variation D2RL and the baselines on a suite of OpenAI-Gym environments. We tabulate effective ranks of the feature matrices (learned weight matrices of the penultimate layer of the networks) for D2RL and a normal MLP, with the TD3 algorithm after 1M interactions, corresponding to Fig. 4. The value is the average over 5 random seeds rounded off to the nearest integer. A loss in rank of the matrix of weights of the penultimate layers (corresponding to learning features) leads to a loss in effectively expressivity of the network, and has been shown to correlate with poor performance (Kumar et al., 2020). Higher is better.

across environments. For this computation, we use the same formula of $srank_\delta(\Phi)$ from (Kumar et al., 2020), where $\Phi$ is the learned weight matrix of the penultimate layer of the nwtwork (i.e. the feature matrix). Kumar et al. (2020) showed that lower effective rank of the feature matrix correlates negatively with performance, and we believe this might help explain some of the empirical observations of better performance of D2RL. Using skip connections as in D2RL leads to higher effective rank feature matrices compared to standard MLPs used in deep RL, as can be seen from Table 3.

**The sample efficiency and asymptotic performance of D2RL scale to complex robotic manipulation and locomotion environments.** Additionally, we consider some challenging manipulation and locomotion environments with different robots, the details of which are discussed below:

*Fetch-{Reach, Pick, Push, Slide}*: There are four environments, where a Fetch robot is tasked with solving the tasks of reaching a goal location, picking an object and placing it at a goal location, pushing a puck to a goal location, and sliding a puck to a goal location. The the Fetch-Slide environment, it is ensured that sliding occurs instead of pusing because the goal location is beyong the end-effector's reach. The observations to the agent consist of proprioceptive state features and the action space is the (x,y,z) position of the end-effector and the distance between the grippers.

*Jaco-Reach*: A Jaco robot arm with a three finger gripper is tasked with reaching a location location indicated by a red brick. The observations to the agent consist only of proprioceptive state features and the arm is joint torque controlled.

*Ant-Maze*: An Ant robot with four legs is tasked with navigating a U-shaped maze while being joint torque controlled. This is a challenging locomotion environment with a temporally-extended task, that requires the agent to move around the maze to reach the goal.

*Baxter-Block Join and Lift*: One arm of a Baxter robot with two fingers must be controlled to grasp a block, join it to another block and lift the combination above a certain goal height. The observations to the agent consist of proprioceptive state features and the action space is the (x,y,z) position of the end-effector and the distance between the grippers.

For the **Fetch**-{Reach, Pick, Push, Slide} environments, we consider the HER (Andrychowicz et al., 2017) algorithm (with DDPG (Qiu et al., 2019)) trained with sparse rewards that was shown to achieve state-of-the-art results in these environments. The plots for **Fetch**- Pick and **Fetch**-Push are in the Appendix, sue to space constraint here. In addition, we also show results with SAC on **Fetch**-Reach and **Fetch**-Slide trained using a SAC agent. For **Ant**-Maze, we consider the hierarchical RL algorithm HIRO (Nachum et al., 2018) that was shown to be successful in this very long horizon task. For **Jaco**-Reach and **Baxter**-Block Join and Lift, we consider the default SAC algorithm released with the environment codebase https://github.com/clvrai/furniture

The results are summarized in Fig. 7, where we see that the proposed D2RL modification converges to higher episodic rewards and converges significantly faster in most environments. By performing a wide range of experiments on challenging robotics environments, we further notice significantly better sample efficiency on all environments which suggests the wide generality and applicability of D2RL. Interestingly, we also observe in 5d that SAC is unable to train the agent to perform the Jaco-Reach task in 3M environment steps, while SAC trained with D2RL policy and $Q$-networks is able to succesfully train an agent and starts outperforming the SAC baseline as early as 1M

environment steps. This shows how crucial parameterization is in some environments as a simple 2-layer MLPs may not be sufficiently expressive or optimization using deeper network architectures may be necessary to solve such environments.

## 5.3 COMPARISONS WITH RESNET

Table 1 tabulates experiments with a ResNet-like MLP which utilizes residual connections instead of dense connections He et al. (2016a). We experiment with images using a base CURL agent Srinivas et al. (2020), and directly from proprioceptive features using a base SAC agent Haarnoja et al. (2018) on the DM Control Suite. We see that using D2RL clearly outperforms the ResNet agent on most tasks from images and state features. Residual connections add the features of a previous layer to the current layer instead of concatenating them (as done in D2RL). The same hyperparameters are used as D2RL for all experiments.

## 5.4 ABLATION STUDIES

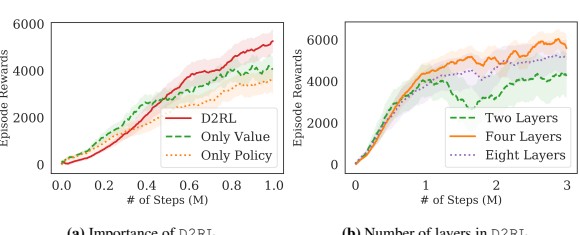

**(a)** Importance of D2RL      **(b)** Number of layers in D2RL

**Figure 6:** Ablation studies with a SAC agent on the Ant-v2 env in the Open AI Gym suite. It is evident that the D2RL policy architecture applied to both the policy and Q-value networks achieves higher rewards than being applied to either just the policy or just the value network. Also, in (b) deeper D2RL networks perform better, in contrast to vanilla MLP networks in Fig. 2

In this section, we look to answer the various components of using D2RL. We first analyze how the agent performs when only the policy or the value functions are parameterized as D2RL, while the other one is a vanilla 2-layer MLP. The results for training an SAC agent on Ant-v2 are present in Fig. 6a, where we see that parameterizing both the networks as D2RL significantly outperforms when only one of the two use D2RL. However, one noteworthy observation can be made that when only the value functions are parameterized using D2RL, the agent significantly outperforms the variant where only the policy is parameterized using a D2RL. This suggests that it may be more important to parameterize the value function, but more research is required to give a more conclusive statement.

Similarly we train the same agent but instead vary the number of layers used while parameterizing the policies and value functions using D2RL. The results in Fig. 6b show that even when 8 layer D2RL is used, the results are only moderately worse that when using 4 layers, even though it has twice the depth and therefore twice as many parameters. These results are notably different from the results in Fig. 2, where as we increase the number of layers for vanilla MLPs to be greater than 2, we see a worsening results. The difference suggests that by using D2RL we are able to circumvent the issue of DPI that may hinder the performance for vanilla MLPs, as we postulated.

## 6 DISCUSSION, FUTURE WORK, AND CONCLUSION

In this paper, we investigated the effect of building better inductive biases into the architectures of the function approximators in deep reinforcement learning. We first looked into the effect of varying the number of layers to parameterize policies and value functions, and how the performance deteriorates as the number of layers increase. To overcome this problem, we proposed a generally applicable solution that significantly improves the sample efficiency of the state-of-the-art DRL baselines over a variety of manipulation, and locomotion environments with different robots, from both states and images. Studying the effect of network architectures has been long explored in computer vision and deep learning, and its benefits on performance have been well established. The effect of the network architectures, however, have not yet been studied in DRL and robotics. Improving the network architectures for a variety of popular standard actor-critic algorithms demonstrates the importance of building better inductive biases in the network paramterization such that we can improve the performance for the otherwise identical algorithms. In future work, we are interested in building better network architectures and further improving the underlying algorithms for robotic learning.

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

## A  PYTORCH CODE

```
1  import torch
2  import torch.nn as nn
3  import torch.nn.functional as F
4
5  LOG_SIG_MAX = 2
6  LOG_SIG_MIN = -5
7
8  class Policy(nn.Module):
9      def __init__(self, num_inputs, num_actions, hidden_dim, action_space=
       None):
10         super(Policy, self).__init__()
11         in_dim = hidden_dim+num_inputs
12         self.linear1 = nn.Linear(num_inputs, hidden_dim)
13         self.linear2 = nn.Linear(in_dim, hidden_dim)
14         self.linear3 = nn.Linear(in_dim, hidden_dim)
15         self.linear4 = nn.Linear(in_dim, hidden_dim)
16         self.mean_linear = nn.Linear(hidden_dim, num_actions)
17         self.log_std_linear = nn.Linear(hidden_dim, num_actions)
18
19     def forward(self, state):
20         x = F.relu(self.linear1(state))
21         x = torch.cat([x, state], dim=1)
22         x = F.relu(self.linear2(x))
23         x = torch.cat([x, state], dim=1)
24         x = F.relu(self.linear3(x))
25         x = torch.cat([x, state], dim=1)
26         x = F.relu(self.linear4(x))
27
28         mean = self.mean_linear(x)
29         log_std = self.log_std_linear(x)
30         log_std = torch.clamp(log_std, min=LOG_SIG_MIN, max=LOG_SIG_MAX)
31         return mean, log_std
32
33  class QNetwork(nn.Module):
34     def __init__(self, num_inputs, num_actions, hidden_dim, num_layers):
35         super(QNetwork, self).__init__()
36
37         in_dim = num_inputs+num_actions+hidden_dim
38         self.l1_1 = nn.Linear(num_inputs+num_actions, hidden_dim)
39         self.l1_2 = nn.Linear(in_dim, hidden_dim)
40         self.l1_3 = nn.Linear(in_dim, hidden_dim)
41         self.l1_4 = nn.Linear(in_dim, hidden_dim)
42
43         self.out1 = nn.Linear(hidden_dim, 1)
44
45     def forward(self, state, action):
46         xu = torch.cat([state, action], dim=1)
47         x1 = F.relu(self.l1_1(xu))
48         x1 = torch.cat([x1, xu], dim=1)
49         x1 = F.relu(self.l1_2(x1))
50         x1 = torch.cat([x1, xu], dim=1)
51         x1 = F.relu(self.l1_3(x1))
52         x1 = torch.cat([x1, xu], dim=1)
53         x1 = F.relu(self.l1_4(x1))
54
55         x1 = self.out1(x1)
56         return x1
```

PyTorch code for a stochastic SAC policy and $Q$-network (Haarnoja et al., 2018). The code provided can simply replace the policy and $Q$-network for any current SAC implementation, or be adopted for other actor-critic algorithms such as TD3 (Fujimoto et al., 2018) or DDPG (Qiu et al., 2019).

## B  ADDITIONAL EXPERIMENTS

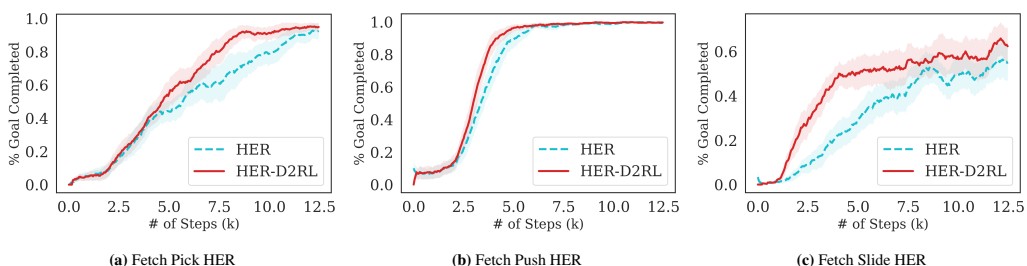

(a) Fetch Pick HER    (b) Fetch Push HER    (c) Fetch Slide HER

**Figure 7:** Complete set of experiments with HER (Andrychowicz et al., 2017) and DDPG (Qiu et al., 2019) on the Fetch robot. Using `D2RL` continues to outperform the baseline on the three environments considered.

## C  HYPERPARAMETERS AND ENVIRONMENT DETAILS

**Table 4:** Hyperparameters used for SAC in all the experiments. The hyperparameter values are kept the same across SAC and SAC-`D2RL` baselines in all the environments.

| Hyperparameter | Value |
|---|---|
| Hidden units (MLP) | 256 for Gym |
|  | 1024 for DM Control (Yarats et al., 2019) |
| Evaluation episodes | 10 |
| Optimizer | Adam |
| Learning rate $(\pi_\psi, Q_\phi)$ | $3e-4$ |
| Learning rate $(\alpha)$ | $1e-4$ |
| MiniBatch Size | 256 |
| Non-linearity | ReLU |
| Discount $\gamma$ | .99 |
| Initial temperature | 0.1 |

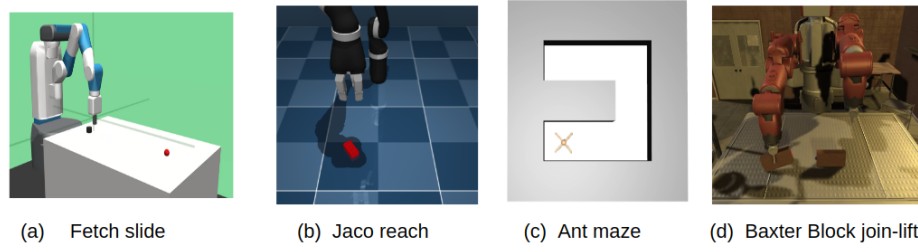

(a)   Fetch slide    (b)   Jaco reach    (c)   Ant maze    (d)   Baxter Block join-lift

**Figure 8:** Illustrations of some of the challenging robotic control environments used for our experiments. In *Fetch slide*, a Fetch robot arm with one finger must be controlled to slide a puck to a goal location. In *Jacko reach*, a Jaco robot with a three finger gripper must be controlled to reach the red brick. In *Ant maze*, an Ant with four legs must be controlled to navigate a maze. In *Baxter block join-lift*, one arm of a Baxter robot with a two finger gripper must be controlled to join two blocks and lift the combination above a certain height.

**Table 5:** Hyperparameters used for DMControl CURL experiments. The hyperparameter values are kept the same across CURL, CURL-`D2RL`, and CURL-ResNet baselines.

| Hyperparameter | Value |
|---|---|
| Random crop | True |
| Observation rendering | $(100, 100)$ |
| Observation downsampling | $(84, 84)$ |
| Replay buffer size | 100000 |
| Initial steps | 1000 |
| Stacked frames | 3 |
| Action repeat | 2 finger, spin; walker, walk |
| | 8 cartpole, swingup |
| | 4 otherwise |
| Hidden units (MLP) | 1024 (Yarats et al., 2019) |
| Evaluation episodes | 10 |
| Optimizer | Adam |
| $(\beta_1, \beta_2) \rightarrow (f_\theta, \pi_\psi, Q_\phi)$ | $(.9, .999)$ |
| $(\beta_1, \beta_2) \rightarrow (\alpha)$ | $(.5, .999)$ |
| Learning rate $(f_\theta, \pi_\psi, Q_\phi)$ | $3e-4$ |
| Learning rate $(\alpha)$ | $1e-4$ |
| Batch Size | 512 (cheetah), 128 (rest) |
| $Q$ function EMA $\tau$ | 0.01 |
| Critic target update freq | 2 |
| Convolutional layers | 4 |
| Number of filters | 32 |
| Non-linearity | ReLU |
| Encoder EMA $\tau$ | 0.05 |
| Latent dimension | 50 |
| Discount $\gamma$ | .99 |
| Initial temperature | 0.1 |

**Table 6:** Consolidated details of all the environments used in our experiments

| Environment | Type | Controller | Inputs | Action dim. | Input dim. |
|---|---|---|---|---|---|
| Gym Cheetah | Locomotion | Joint torque | States | 6 | 17 |
| Gym Hopper | Locomotion | Joint torque | States | 3 | 11 |
| Gym Humanoid | Locomotion | Joint torque | States | 17 | 376 |
| Gym Walker | Locomotion | Joint torque | States | 6 | 17 |
| Gym Ant | Locomotion | Joint torque | States | 8 | 111 |
| Finger, Spin | Classical control | Joint torque | States/Images | 2 | 9 / 84x84x3 |
| Cartpole, Swing | Classical control | Joint torque | States/Images | 1 | 5 / 84x84x3 |
| Reacher, Easy | Classical control | Joint torque | States/Images | 2 | 6 / 84x84x3 |
| Cheetah, Run | Locomotion | Joint torque | States/Images | 6 | 17 / 84x84x3 |
| Walker, Walk | Locomotion | Joint torque | States/Images | 6 | 24 / 84x84x3 |
| Ball in a Cup, Catch | Manipulation | Joint torque | States/Images | 2 | 8 / 84x84x3 |
| Fetch Reach | Manipulation | EE position | States | 4 | 10 |
| Fetch Pick and Place | Manipulation | EE position | States | 4 | 25 |
| Fetch Push | Manipulation | EE position | States | 4 | 25 |
| Fetch Slide | Manipulation | EE position | States | 3 | 25 |
| Jaco Reach | Manipulation | Joint torque | States | 9 | 45 |
| Baxter JoinLift | Manipulation | Joint torque | States | 15 | 42 |
| Ant Maze | Locomotion | Joint torque | States | 8 | 30 |

