# OpenReview forum: "D2RL: Deep Dense Architectures in Reinforcement Learning"
_ICLR.cc/2021/Conference — Reject_

### Official Review · AnonReviewer2 · 2020-10-19
**Official Blind Review #2**

**Rating:** 4
**Confidence:** 4

**Review:**

##########################################################################

**Summary**:

This paper investigates the effect of different network architecture in the context of reinforcement learning. It shows that by appending the input to each mid-layer's output, one can use a deeper network to get better learning performance. The idea is very similar to the residual connection or the skip connection. And I don't see too much novelty in applying such an idea in RL settings.


##########################################################################

**Strengths**:

The paper is well-written and provides PyTorch code that is easy to read and understand.

It experiments across a wide range of environments and tasks.

##########################################################################

**Weaknesses**:

The paper is investigating the use of skip connection in deeper networks in RL. The skip connection can be a summation operation like ResNet or a concatenation operation like DenseNet. This paper uses concatenation. Such an idea is not new to the learning community. There is nothing specific in RL that prevents one from using such standard techniques in the networks either. It is common to use skip connections in a deep network, even in RL [1, 2, 3]. The novelty of the paper is limited. And I would like to see a more systematic and thorough analysis of why this is a good choice people should choose and how it compares to other ways of skip connections, etc.


Figures 3, 4, 5 show the results on a shallow (2 layers) network and a relatively deep (4 layers) network with skip connections. It is not clear whether the effect solely comes from a deep network or the skip connection. Even though Figure 2 shows that a deep network does not perform well in Ant-v2, this might not hold true in the other environments. Hence, I would like to see the learning curves of a deep network without skip connections. Also, since the network becomes deep and RL is sensitive to hyperparameters, it makes sense to tune the hyperparameters as well. We should compare the performance of two scenarios when each of them is best tuned.

Concatenating the input to each mid-layer also makes each layer wider. What about using addition instead of concatenation? One can simply use a residual connection in each layer until the last one. Each layer will be `y=x+f(x)`, which is fairly common in many deep networks. How does it perform? And what about using the deep network with the same number of parameters without a skip connection?


The paper only shows the results with one kind of network width. As shown in [4], network architecture (both the width and depth) has a significant effect on the outcomes. I would like to see the effectiveness of D2RL on networks with a few different widths.



Missing details:
* Reward structure for the environments.
* It is not clear how many skip connections are added in CURL. More details should be provided.

[1] Espeholt, Lasse, et al. "Impala: Scalable distributed deep-rl with importance weighted actor-learner architectures." arXiv preprint arXiv:1802.01561 (2018).

[2] Gupta, Saurabh, et al. "Cognitive mapping and planning for visual navigation." Proceedings of the IEEE Conference on Computer Vision and Pattern Recognition. 2017.

[3]: Finn, Chelsea, and Sergey Levine. "Deep visual foresight for planning robot motion." 2017 IEEE International Conference on Robotics and Automation (ICRA). IEEE, 2017.

[4] Henderson, Peter, et al. "Deep reinforcement learning that matters." arXiv preprint arXiv:1709.06560 (2017).

---

> ### Author Response · Authors · 2020-11-18
> **Author response (2/2): Clarified learning curves and hyperparameter setting, added detailed comparisons to ResNet**
>
>
> **Missing details:**
>
> **Reward structure for the environments.**
> Thank you for pointing this out. We consider the standard reward functions for all the environments in Open AI Gym, DM Control Suite, the Ant Environments (for HIRO), and IKEA Furniture Assembly, and perform no further reward shaping. The respective papers are cited for better reference to the environments.
>
> **It is not clear how many skip connections are added in CURL. More details should be provided.**
>
> For a base CURL agent, and when working with images, we simply concatenate the output of the CNN layer for the policy and the value function, along with the action for the value function to each intermediate layer. This architecture is similar to the one in SkipVAE where they concatenate the embedding features to the intermediate layers (Dieng et al., 2019).
>
>
> [1] He, K., Zhang, X., Ren, S. and Sun, J., 2016. Deep residual learning for image recognition. In Proceedings of the IEEE conference on computer vision and pattern recognition (pp. 770-778).

---

> ### Author Response · Authors · 2020-11-18
> **Author response (1/2): Clarified learning curves and hyperparameter setting, added detailed comparisons to ResNet**
>
> Thank you for the detailed review and list of concerns. The main concerns pointed out in the review are clarification about learning curves for the normal network baseline without skip connections, clarification about varying network width and hyperparameter tuning, and the necessity for new experiments with the ResNet baseline. We have added new ResNet baselines in Table 1, and clarified the other concerns pointwise below.
>
> In light of these revisions and updated results, we request the reviewer to kindly look at our responses and let us know if anything is unclear, or if we can improve the paper further.
>
> Our revised manuscript contains major revisions highlighted in blue. In the points below, we first paraphrase text from the review in bold and follow it with our response in plain text.
>
>
> **Figures 3, 4, 5 show the results on a shallow (2 layers) network and a relatively deep (4 layers) network with skip connections. It is not clear whether the effect solely comes from a deep network or the skip connection. Even though Figure 2 shows that a deep network does not perform well in Ant-v2, this might not hold true in the other environments. Hence, I would like to see the learning curves of a deep network without skip connections.**
>
> Apologies if we did not understand this question clearly. But we already provide learning curves in Figures 3, 4, and 5 of the paper. In the Figures, the baselines without "-D2RL" are deep networks without skip connections. We note that based on the ResNet results (that have the same depth as D2RL), adding depth does not yield good results - since the Resnet results are much worse than D2RL, the effect does not come from deeper networks, and indeed comes from skip connections.
>
> In each of the Open AI gym environments the performance of increasing the depth of the networks significantly deteriorates the performance of the agent. We simply omit the other environments for brevity and show results for Ant-v2 as an example of the expected trend. We are limited by the amount of compute and time during the rebuttal phase, we will add these figures for other environments to the supplementary in the following weeks.
>
>
> **Also, since the network becomes deep and RL is sensitive to hyperparameters, it makes sense to tune the hyperparameters as well. We should compare the performance of two scenarios when each of them is best tuned.**
>
> Thank you for this suggestion. We kept the hyperparameters fixed for all algorithms on all environments, and did not tune them to ensure fair comparison - as some algorithms might require more tuning to obtain optimal performance. For all experiments, we used the exact same set of hyperparameters as were proposed in the original papers. Keeping all hyperaprameters fixed is necessary to isolate effects of the architectural changes (e.g. the ResNet paper [1] had ablations for different layers in the model keeping everything else the same) For ex: we use the exact same hyperparameters as the official code release of the TD3 paper: https://github.com/sfujim/TD3. It is more likely that the agents using D2RL can be tuned further since it's possible that the default parameters for an agent are for the vanilla MLP case and not for D2RL. But to ensure fair comparisons between the baselines, we use the default parameters for all our experiments.
>
> **Concatenating the input to each mid-layer also makes each layer wider. What about using addition instead of concatenation? One can simply use a residual connection in each layer until the last one. Each layer will be y=x+f(x), which is fairly common in many deep networks.**
>
> We have now added results for comparison with Resnet architecture, as suggested. This corresponds to addition at each layer y=x+f(x), instead of concatenation. This is in Table 1 of the revised paper. We have added results for training from both states and images, with two different algorithms, CURL and SAC. We see that the results with D2RL are consistently better, especially with image observations than the ResNet variant, and also significantly more sample efficient than the base SAC agent in 100k steps on the DM Control Suite environments.
>
>
>
> **The paper only shows the results with one kind of network width. As shown in [4], network architecture (both the width and depth) has a significant effect on the outcomes. I would like to see the effectiveness of D2RL on networks with a few different widths.**
>
> Yes, we actually had different widths of networks. In the OpenAI Gym environment results, the width of all the networks are 256 (as reported in the paper), and for the DM Control Suite experiments, the width of the networks are all 1024 (as reported in Yarats et al., 2018).

---

> ### Author Response · Authors · 2020-11-22
> **Discussion**
>
> Kindly let us know if our response below addressed your concerns. We will be happy to answer if there are additional issues/questions.

---

> > ### Comment · AnonReviewer2 · 2020-11-24
> > **Please fix the fonts in the tables**
> >
> > Thanks for your response.
> >
> > Just a few more comments:
> >
> > * Please fix the "bold fonts" in the tables. There is no point to highlight the results of all methods altogether.
> >
> > * I am a bit confused by the response on why not **all** the experiment curves are provided for other Gym environments. While I understand the limitations of computing, the explanation is kind of self-conflicting. On one hand, the authors claim that the performance of increasing network depth significantly deteriorates the performance for other Gym environments, which implies that the authors have run those experiments and seen the results on those environments. On the other hand, the authors also say they are limited by computing and can only provide those curves in the following weeks. And I don't think we can omit environments just for the sake of brevity in a research paper.
> >
> > * Can you also provide the learning curve of ResNet-style architectures in Figure 2 (as pointed out by R4)? It's more clear how addition vs concatenation affect learning in a figure with 1M steps. We can see if the learning is stable, how big the variance is along with the entire training process, not just on two checkpoints (100k and 500k steps as provided in the paper).
> >
> > * If adding the depth of the network is not the reason for performance gain, it seems that it is the concatenation that is playing the role here. So how do the following compare to each other: (1) D2RL with 4 layers, (2) D2RL with 2 layers, (3) normal setup with 2 layers? Does (2) also perform better than (3)?

---

> > > ### Author Response · Authors · 2020-11-24
> > > **Thank you for your reply. We have responded to the additional comments below.**
> > >
> > > Thank you for acknowledging the new results we have added in the paper.
> > >
> > > We have removed boldface from Table 1 where all the methods have similar performance. We hope this addresses the concern. Kindly let us know if we misunderstood your point regarding this.
> > >
> > > We have updated the paper by moving the missing Gym plot (Hopper with SAC) from the appendix to the main paper.
> > >
> > > Regarding plots for the DM Control environments, thank you for suggesting this. In addition to the 100k and 500k results, we will provide all the training curves for these environments with D2RL and ResNet variants in the project website https://sites.google.com/view/d2rl-anonymous/home (linked to the abstract now) since the author response is ending tonight and we won't be able to update the plots in the appendix by tonight. Thank you for your understanding.
> > >
> > > We had plots for (1) D2RL with 4 layers, (2) D2RL with 2 layers, (3) normal setup with 2 layers in the paper (Figure 2 and Figure 6 (b)). For comparison, the values at 500k for (3) and (2) are:
> > >
> > > walker-2layers:3003 $\pm$ 536
> > >
> > > walker-d2rl-2layers:3156 $\pm$ 430
> > >
> > > hopper-2layers: 2321 $\pm$ 492
> > >
> > > hopper-d2rl-2layers: 2560 $\pm$ 531
> > >
> > >
> > > We have also explained in the ablation studies section 5.2, the effects of varying the depth of the networks used for parametertization.
> > >
> > > We would be grateful if you kindly let us know if there is anything else we should clarify for a revised positive assessment of our paper and the rating score.

---

> > > > ### Comment · AnonReviewer2 · 2020-11-24
> > > > **Where are the results for walker and hopper in the paper?**
> > > >
> > > > Thanks for the timely reply.
> > > >
> > > > I didn't see these results in the paper:  walker-2layers , walker-d2rl-2layers,  hopper-2layers, hopper-d2rl-2layers. Are these new results that have not been put in the paper? Or did I miss something from the paper?
> > > >
> > > > From Figure 2 and Figure 6(b), it shows that D2RL-2layers performs worse than normal-2layers, what would be your explanation/insight for this?

---

> > > > > ### Author Response · Authors · 2020-11-24
> > > > > **Clarification: new results indeed**
> > > > >
> > > > > The results with D2RL-2 layers and MLP-2 layers are similar in ant (in Figure 2 and Figure 6b), and also similar in the walker and hopper (provided in the previous comment). These are indeed new results that we will add into the final version of the paper, as well as corresponding plots. We also add a full discussion related to this in the appendix of the draft in the final version of the paper.
> > > > >
> > > > > D2RL with 2 layers does not benefit from added depth, and simply concatenates the input to each layer. But as the number of layers in the networks increase, we see a steep decline in performance for the vanilla MLPs (Figure 2). Using dense connections, D2RL is able to overcome this and significantly outperform the vanilla MLP baseline. The reason for the improved performance is due to increase in hidden layers, which is made possible by the dense connections (this is also supported by the rank collapse experiments).

---

### Official Review · AnonReviewer1 · 2020-10-29
**Resnet should be compared**

**Rating:** 4
**Confidence:** 4

**Review:**

This paper proposes a deep neural net structure for deep reinforcement learning methods (e.g., SAC) to replace the original fully-connected layers, by concatenating the state input into every hidden layers. The authors conduct experiments on OpenAI gym and MuJoCo environments and show that the proposed structure can further improve the performance of SAC or CURL.

Strong points:
1. The authors propose a method by concatenating the state features to every layer of the neural net to improve the performance of RL algorithm. The proposed method seems to have overcome the issue that purely adding more layers of fully-connected network can even harm the performance.

Weak points:
1. The biggest issue of this work is that the proposed method, regardless of the activation function, is similar to a special version of resnet. Stacking residual layers can make it possible to have skip connections from every layer to any layer after it. Thus, resnet has included the connection directly from feature input to each layer. Therefore, this method seems to lack enough technical innovation.
2. The authors should compare with other types of neural net structures that aim to solve the "depth" problem. At least, resnet should be compared.


Minor comments:
Will this network structure also work for supervised learning problems? It seems this structure is independent from the RL setting.

---

> ### Author Response · Authors · 2020-11-18
> **Author response: Added detailed comparisons to ResNet, included analysis for intuition of observed benefits**
>
> Thank you for the detailed review of our paper. The main concerns pointed out in the review are the necessity for comparison with ResNet and clarification of how D2RL is different from ResNet. We have added a new ResNet baseline in Table 3, and elaborated in the response below that while ResNet involves addition at each layer, D2RL performs concatenation.
>
> In light of these revisions and updated results, we request the reviewer to kindly look at our responses and let us know if anything is unclear, or if we can improve the paper further.
>
> Our revised manuscript contains major revisions highlighted in blue. In the points below, we first paraphrase text from the review in bold and follow it with our response in plain text.
>
> **The biggest issue of this work is that the proposed method, regardless of the activation function, is similar to a special version of resnet. Stacking residual layers can make it possible to have skip connections from every layer to any layer after it. Thus, resnet has included the connection directly from feature input to each layer. Therefore, this method seems to lack enough technical innovation.**
>
> We would like to respectfully point out that the proposed approach of adding skip connections, although similar to residual connections, is quite different empirically. Empirically, in comparison to ResNet, we see that the proposed D2RL performs much better in both image-based and state-based environments. These results are in Table 1 of the revised paper.
>
>
> **The authors should compare with other types of neural net structures that aim to solve the "depth" problem. At least, resnet should be compared.**
>
> We have now added results for comparison with Resnet architecture, which is one of the popular variants. This corresponds to addition at each layer y=x+f(x), instead of concatenation. This is in Table 1 of the revised paper. We have added results for training from both states and images, with two different algorithms, CURL and SAC. We see that the results with D2RL are consistently better, especially with image observations than the ResNet variant, and also significantly more sample efficient than the base SAC agent in 100k steps on the DM Control Suite environments.

---

> ### Author Response · Authors · 2020-11-22
> **Discussion**
>
> Kindly let us know if our response below addressed your concerns. We will be happy to answer if there are additional issues/questions.

---

### Official Review · AnonReviewer3 · 2020-10-30
**novel and effective architecture for deep reinforcement learning**

**Rating:** 8
**Confidence:** 3

**Review:**

This submission takes inspiration from work on deep learning architectures for visual tasks in order to make targeted model changes to deep reinforcement learning models. The authors show that by including “dense connections” (concatenating the state or state-action pair to the input of each hidden layer of the network) they are able to successfully train deeper networks.

The main idea behind the work is simple but effective, and their model surpasses the most of the presented benchmarks. The paper is also well written and presents a thorough set of experiments, making it a good submission for ICLR.

Positives:
* The main idea behind the architecture is fairly simple and the explanation is grounded in previous architectures (specifically densenet), making the experiments quite easy to understand.
* The authors evaluate their method on a diver set of tasks, and their model outperforms the benchmark for the majority of tested conditions

Concerns and Questions:
* The results comparing the ResNet style architecture with the DenseNet style architecture are interesting, particularly because the ResNet architecture does not see the same benefit. An explanation of how to interpret this result would be helpful to readers (ie why does a residual connection in this setting not help with DPI?).
* It is unclear why the authors chose to use a 4 layer D2RL. It looks like an experiment was done in 6b varying the number of layers, but perhaps introducing this earlier (ie as a direct comparison to Figure 2) would make this choice more clear.

---

> ### Author Response · Authors · 2020-11-18
> **Author response**
>
> Thank you for the detailed review and very encouraging comments about the paper.
>
> Our revised manuscript contains major revisions highlighted in blue. In the points below, we first paraphrase text from the review in bold and follow it with our response in plain text.
>
>
> **The results comparing the ResNet style architecture with the DenseNet style architecture are interesting, particularly because the ResNet architecture does not see the same benefit. An explanation of how to interpret this result would be helpful to readers (ie why does a residual connection in this setting not help with DPI?).**
>
> Thank you for pointing this out. We have updated the comparisons with ResNet with more results in Table 1.  Since we submitted this paper to ICLR, there has been one paper released on arXiv that talks of implicit under-parameterization in standard MLPs used for function approximation in RL https://arxiv.org/pdf/2010.14498.pdf The key result here is that the penultimate layer of the policy and value networks that corresponds to the learned feature matrix suffers a rank collapse i.e. it's rank is much less than a full rank matrix. We believe this might help explain why D2RL performs better than standard MLPs for RL. Since we feed in the input to each layer of the network, rank collapse is significantly alleviated. We empirically verify this in our experiments by measuring the rank of the feature matrix with torch.svd, and update the discussion in the revised paper in section 5.2 and Table 3.
>
>
> **It is unclear why the authors chose to use a 4 layer D2RL. It looks like an experiment was done in 6b varying the number of layers, but perhaps introducing this earlier (ie as a direct comparison to Figure 2) would make this choice more clear.**
>
> We provide an ablation in Figure 6b where we discuss the importance of the number of layers. We see that when D2RL has 8 layers, then the learning is also harmed as it's possible that the network has too many parameters to optimize with sample efficiency. We see that the tradeoff between the two is best solved when the number of layers is 4.

---

> > ### Comment · AnonReviewer3 · 2020-11-24
> > **Follow Up**
> >
> > Thank you for the clarifications, especially the discussion about rank collapse with regards to the ResNet results. I have no unanswered questions.

---

### Official Review · AnonReviewer4 · 2020-10-30
**The work does empirically improve performance on a range of fully observable benchmarks. But lacks analysis and real novelty**

**Rating:** 5
**Confidence:** 4

**Review:**

In this work, the authors propose a neural network architecture that concatenates the input state with hidden state activations over multiple layers in order to train deeper networks in an RL setting. Whilst the work does improve over standard MLP in this setting, is seems like an incremental work that lacks real novelty.

The idea of residual connections or concatenation to improve stability of networks is not a new one. Although there is nothing technically wrong with this paper and there is an improvement over a vanilla network, I do not feel the work is enough for a publication at ICLR, the work is not novel enough and the authors should focus on bigger steps rather than incremental work.

The following changes would be required for me to up my rating:
1. More ablations, particularly vs. resnet architectures, it would be good to see figure 2 with a resnet comparison.
2. Analysis of why the standard MLP case fails, is the weight activation suitable, are there vanishing gradients? I find the discussion about DPI a bit hand-wavey.
3. Comparisons on other environments such a Atari and even partially observable environments (DM-Lab, Habitat...)


As many of these changes are out of scope for a rebuttal, I would suggest that this publication is not ready for a venue such as ICLR and should either be greatly extended to a larger suite of scenarios such as Atari or submitted to more suitable conference.

Update:
I thank the authors for their significant updates to the paper.  Given the extended effort made by the authors, I am willing to raise my score to 5. My conclusion however remains the same, this work is not a significant advancement that we would expect to see at a conference such as ICLR.

---

> ### Author Response · Authors · 2020-11-18
> **Author response: Added more experiments, included analysis for intuition of observed benefits**
>
> Thank you for the detailed review and list of concerns. The main concerns pointed out in the review are the
>
> **necessity for more experiments**  - We have added a new ResNet baselines in Table 3
>
> **interpretation of why we observe benefits with D2RL** - We have added a discussion section based on implicit under parameterization that helps explain the better performance of D2RL compared to using a normal MLP for function approximation in RL
>
> **evaluation on Atari** - We are currently running experiments for Atari which we will update when they finish running.
>
>
>
> Our new results show that the ResNet variant does not perform as well as D2RL across tasks and algorithms (Table 1). In light of these revisions and updated results, we request the reviewer to kindly look at our responses and let us know if anything is unclear, or if we can improve the paper further.
>
> Our revised manuscript contains major revisions highlighted in blue. In the points below, we first paraphrase text from the review in bold and follow it with our response in plain text.
>
>
> **More ablations, particularly vs. resnet architectures**
>
> We have now added results for comparison with Resnet architecture, as suggested. This corresponds to addition at each layer y=x+f(x), instead of concatenation. We see that the results with D2RL are consistently better, especially with image observations than the ResNet variant, and also significantly more sample efficient than the base SAC agent in 100k steps on the DM Control Suite environments. This is in Table 1 of the revised paper. We have added results for training from both states and images, with two different algorithms, CURL and SAC.
>
>
> **Analysis of why the standard MLP case fails, is the weight activation suitable, are there vanishing gradients? I find the discussion about DPI a bit hand-wavy.**
>
> Thank you for pointing this out. Since submission, there has been concurrent work [1] that talks of implicit under-parameterization in standard MLPs used for function approximation in RL. The key result here is that the penultimate layer of the policy and value networks that corresponds to the learned feature matrix suffers a rank collapse i.e. its rank is much less than a full rank matrix. We believe this might help explain why D2RL performs better than standard MLPs for RL. Since we feed in the input to each layer of the network, rank collapse is significantly alleviated. We empirically verify this in our experiments by measuring the rank of the feature matrix with torch.svd, and update the discussion in the revised paper (Section 5.2 and Table 3).
>
> **Comparisons on other environments such a Atari and even partially observable environments (DM-Lab, Habitat...)**
>
> We are currently running Atari experiments with a Double-DQN model and its D2RL variant. We will update our response here when we obtain the results.
>
> [1] Kumar, A., Agarwal, R., Ghosh, D. and Levine, S., 2020. Implicit Under-Parameterization Inhibits Data-Efficient Deep Reinforcement Learning. arXiv preprint arXiv:2010.14498.

---

### Author Response · Authors · 2020-11-25
**New independent research findings**

We also kindly would like to bring forward an $\textbf{independent}$ research study that cites our anonymous manuscript and uses D2RL and shows important improvements for baseline SAC and TD3 agents on real-world robotic learning tasks (Boney et al., 2020). This is promising signs of adoption for the learning community which indicates the further utility and uses of D2RL by the reinforcement learning and robotic learning communities.

Boney et al., 2020. RealAnt: An Open-Source Low-Cost Quadruped for Research in Real-World Reinforcement Learning, https://arxiv.org/abs/2011.03085.

---

### Decision · Program_Chairs · 2021-01-07
**Final Decision**

**Decision:**

Reject

**Comment:**

The paper shows that replacing fully connected layers by dense layers in the networks used by actors and critiques in RL can improve the results significantly.  The improvements for several RL techniques across several benchmarks are very nice.  That being said, replacing fully connected layers by dense layers is not particularly novel and it is not clear why dense layers instead of resnet layers works better.  The reviewers appreciate the addition of experiments confirming that dense layers work better than resnet layers.  This addresses an important concern of the reviewers.  However, at this point in deep learning, it is well-known that fully connected layers do not work well in general and therefore engineers are expected to use resnet, dense or highway style connections to improve performance when increasing the depth.  The fact that published baselines in OpenAI, TensorFlow and PyTorch do not use those improved networks is one thing, but this does not justify the publication of a paper.  The paper suggests that an RL-specific architecture will be proposed, but at the end of the day what is being proposed is not specifically for RL, but rather the addition of new connections to the inputs similar to the well-known dense architecture to augment fully connected layers in RL.  It is not clear why this works better than resnet connections.  Another alternative that was not considered are highway networks.  To strengthen the contribution of the paper, the authors are encouraged to provide an analysis of the possible approaches and to provide some insights.